# Assessment of Teacher Motivation, Psychometric Properties of the Work Tasks Motivation Scale for Teachers (WTMST) in Spanish Teachers

Julia Criado-Del Rey [1], Iago Portela-Pino [2,*], José Domínguez-Alonso [3] and Margarita Pino-Juste [4]

[1] Department of Psychology, University of Vigo, 36214 Vigo, Spain; jcrirey@uvigo.es
[2] Department of Health Sciences, Isabel I of Castile International University, 09003 Burgos, Spain
[3] Department of Psycho-Socio-Educational Analysis and Intervention, University of Vigo, 36214 Vigo, Spain; jdalonso@uvigo.es
[4] Department of Didactics, School Organization and Research Methods, University of Vigo, 36214 Vigo, Spain; mpino@uvigo.es
* Correspondence: iagoportt92@gmail.com

**Abstract:** The lack of motivation among teachers is of increasing concern. Consequently, identifying motivations for their teaching performance can help to improve the quality and effectiveness of educational systems. The aim of this article was to translate the Work Tasks Motivation Scale for Teachers (WTMST) into Spanish, an instrument aimed at exploring teachers' motivation and analyzing its psychometric properties. A non-probabilistic sample of 369 teachers (71.3% women) with an age measurement of 44.93 years (SD = 9.58) was used. Exploratory factor analysis and confirmatory factor analysis were applied. The AFE suggested a first solution of four factors that explained 80.57% of the variance in the model. Likewise, a new factor analysis was generated with the extraction of a fixed number of factors indicated in the theoretical review, explaining 86.20% of the total variance. It was concluded that the WTMST is a scale that has evidence of validity and reliability, and that can be considered a valuable contribution to the evaluation of teacher motivation regarding specific tasks in Spanish-speaking populations.

**Keywords:** teacher motivation; assessment; quantitative research

## 1. Theoretical Framework

In recent years, there has been a growing interest in the study of teacher motivation. However, there are few studies that analyzed the motivation of teachers related to their regular academic tasks [1–4].

Teachers today are facing a situation of change in which they are asked to assume a different role than the one they are used to [5,6].

The conceptual mastery of the topics or the manner of transmitting the information to the students are no longer so important, but to turn our students into expert-learners, to make them interested in continuous learning and in research, and to offer them all the possible resources to achieve this goal. We have to face new challenges in the face of the numerous and diverse changes that have arisen in the new knowledge society, and education has the task of educating subjects in these new areas [7]. The teacher then becomes a guide, facilitator, and advisor for the acquisition of competencies in students [8,9]. Similarly, Tébar (2003) suggested that the didactic task of the teacher as a transmitter should be limited and that the teaching process should focus on advising and providing the student with the most appropriate resources for each situation [10].

In contemporary education, educators are a central component of academic institutions, they have a pivotal role in presenting and planning an effective academic program [11]. Without teachers there is no teaching; however, the teacher attrition rate is high

worldwide and teacher shortages have been a problem for decades [12]. Results obtained by Sato (2022) showed that 6–12% of teachers leave the profession in the first year and that after 10 years, about 30% had left their teaching job [13]. When we do not have qualified teachers in a school, the quality of teaching decreases and student learning is negatively affected. Research has shown that one of the main causes of leaving the teaching profession is burnout [14] and that burnout is significantly affected by motivation. Moreover, studies have shown that teachers, more than any other professional, suffer from high demotivation towards their work [15,16].

Teachers who have high motivation towards their profession are more involved, publish more, innovate more, enjoy more, and are more satisfied with their work life in general [17–20]. They undertake more continuing education during their professional career [21]. Moreover, teacher motivation is directly related to student motivation [22], directly influencing it.

Considering the very high number of tasks that teachers are obliged to carry out and the different evaluation processes they face on a daily basis (management, head of studies, guidance department, colleagues, parents, and students), it is difficult to accurately identify the motivational process of each of these tasks. Therefore, it would be useful to have a tool that analyses the motivation of teachers towards each of the tasks entrusted to them, because of their involvement in the success of the teaching–learning process. On the other hand, motivational processes need not be uniform and may vary according to the tasks to be performed. Good teachers are expected, among other functions, to be effective in the teaching–learning process, to maintain students' interest, to avoid frustration and failure, to develop positive attitudes, to create a good classroom climate, to mediate between the student and the contents of the subject, and to be up to date in the scientific and pedagogical field [23–26].

The Work Tasks Motivation Scale for Teachers (WTMST) designed by Fernet et al. (2008) [27] allows us to determine the motivational level of teachers based on the principles of self-determination theory [28]. According to Deci and Ryan's Self-Determination Theory (2000) [28], each of the basic modalities of motivation lies between two extremes of a continuum in terms of self-determination.

Following the definitions of Deci and Ryan (2000) [28], amotivation is located at one end of the continuum and corresponds to an absolute lack of motivation, both intrinsic and extrinsic, where the person is not motivated at all. Extrinsic motivation is determined by extrinsic rewards or agents, the behavior is performed to satisfy an external demand or due to the existence of rewards. In introjected motivation, the regulation of behavior still has an external locus of control, the motives for participation in an activity are mainly social recognition, internal pressures, or feelings of guilt. As for identified regulation, a behavior is highly valued and the individual judges it as important, so he will perform it freely even if the activity is not to his liking. Finally, intrinsic motivation can be defined as that related to the need to explore the environment, the curiosity and pleasure experienced when performing an activity, without receiving direct external gratification. The development of the activity itself constitutes the goal and the gratification, also arousing feelings of competence and self-fulfillment. An important aspect of intrinsically motivated behavior is that the interest in the activity and the needs for competence and self-fulfillment persist even after the goal has been achieved.

In short, in amotivation and external regulation, behavior is completely lacking in self-determination; in contrast, in identified regulation and intrinsic motivation, behavior is considered to be completely self-determined (see Figure 1). Within the framework of these varying levels of self-regulation and self-determination, researchers have proposed different composite variables or indicators calculated from the basic modalities of motivation. Thus, autonomous motivation synthesizes intrinsic motivation and identified regulation; similarly, controlled motivation summarizes external and introjected regulation [29–31]. Finally, some authors consider it useful to obtain a global self-determination index, in which all the basic modalities of motivation studied are synthesized [32].

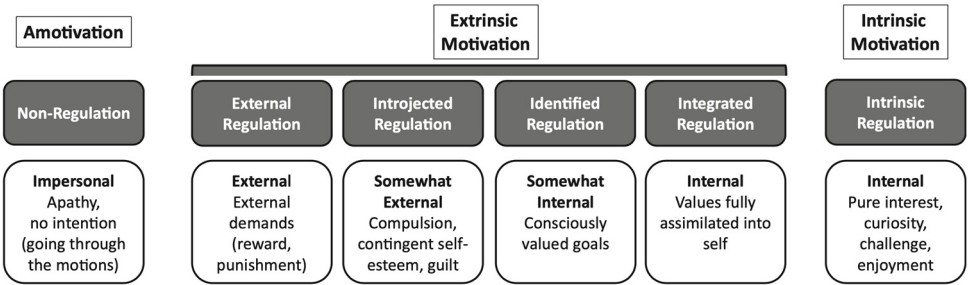

**Figure 1.** Source: Cook and Artino (2016) [33].

There are other instruments for measuring teacher motivation, such as The Intrinsic Motivation Inventory (IMI) by Ryan (1982), [34] but it is not specific for teachers, although it can be adapted to this group. This instrument offers a multidimensional measure of the subjective involvement of the subjects in a specific task. It has been used in different experiments related to intrinsic motivation and self-regulation [34–39].

Another interesting instrument is the scale "The Teacher Motivation and Job Satisfaction Survey" [40], which consists of 12 questions and measures teacher motivation and its relationship with job satisfaction.

In addition, Roth et al. (2007) [41] developed a subscale that assesses four types of teaching motivation within the framework of self-determination theory. It serves to examine teachers' autonomous motivation for teaching and its correlates in teachers and students. The assessment was conducted in a task-specific format, and for each teaching-related task, four responses representing four different types of motivation were assessed: external, introjected, identified, and intrinsic. A subsequent study by Hein et al. (2012) [42] demonstrated the appropriateness of using the instrument for physical education teachers, and thus confirmed the positive relationship between teacher autonomous motivation and student-centered or reproductive teaching styles.

In Spain, teachers Rodríguez et al. (2009) [43] used a teaching motivation questionnaire following the goals theory, in which each teacher assessed the extent to which different reasons explained his or her involvement in the teaching activity.

In the field of teacher motivation in higher education, Visser-Wijnveen et al. (2012) [44] developed and validated a questionnaire based on three already institutionalized questionnaires, but now including the following motivational aspects: efficacy, interest, and effort. The results of the exploratory study showed that after modifications, including the elimination of two of the three efficacy aspects ("efficacy of results" and "efficacy of teaching"), this instrument was sufficiently reliable and valid for use in educational practice and research.

Another measure is the English Teacher Motivation Scale (ETMS), which revealed that English teacher motivation is multidimensional and comprises four main factors: teacher efficacy, school leadership, negative influences, and intrinsic compensation [45].

However, "The Work Tasks Motivation Scale for Teachers" (WTMST) has been chosen as it measures teacher motivation by discriminating between the different tasks they face every day. This 90-item scale is designed to measure five motivational constructs in relation to six professional tasks of the teacher (teaching, preparing classes, evaluating, classroom management, administrative tasks, and complementary tasks). Therefore, the aim of this study was the translation and subsequent identification of the psychometric properties of an instrument to assess teacher motivation towards specific tasks of the WTMST scale [27]. Our aim was to analyze the validity of the instrument for measuring teacher motivation and its internal consistency, and to determine the factorial structure of the scale.

The hypotheses of this study were:

**H1.** *The WTMST scale offers adequate psychometric properties (validity and reliability) for the assessment of teacher motivation towards specific tasks of their profession.*

**H2.** *The five-factor theoretical model of the WTMST scale (intrinsic motivation, identified regulation, introjected regulation, external regulation, and demotivation) represents the best structure for the measurement of the construct in the Spanish teaching population.*

## 2. Method

### 2.1. Participants

The sample used for the two studies was a non-probabilistic convenience sample [46] and consisted of 369 teachers who teach in early childhood, primary, and secondary education in Galicia (Spain). Of these, 71.3% were women (a fact that, far from being a bias in the sample, is consistent with the ordinary patterns of presence of men and women in teaching in Spain) with a mean age of 44.93 years (SD = 9.58). However, this sample was randomly divided into two subsamples: 150 teachers participated in the first study (Gender: 39 males and 111 females; Age: M = 35.39, SD = 4.74) and 219 teachers participated in the second study (Gender: 67 males and 153 females; Age: M = 51.42, SD = 5.88). Thus, it was a non-probabilistic sample of volunteer teachers, with guarantees of randomness and independence.

### 2.2. Instrument

For the adaptation of the Work Tasks Motivation Scale for Teachers (WTMST) to Spanish, the classic backward translation procedure indicated by Muñiz et al. (2013) [47] was followed, which encompasses the following actions:

- Translation of the scale into Spanish from the original English version by three bilingual persons with experience in the field of educational evaluation. These translations were discussed with the research team until a consensus was reached and the first Spanish version was developed.
- An educator with teaching experience in English evaluated the conceptual equivalence, clarity, and contextuality of each of the sentences and answer options of this first version. With the pertinent rectifications, a new version was obtained.
- Consultation was also carried out with educators who are experts in the teacher's own tasks (class preparation, teaching, student assessment, classroom management, administrative and complementary tasks, etc.).
- A second Spanish version was obtained, which was then translated back into English by a native bilingual translator.
- A pilot test was then carried out with 75 teachers (25 teaching in early childhood education, 25 in primary education, and 25 in secondary education) in order to assess the comprehension, time required to complete the questionnaire, clarity of the questions, and adequacy of the answers.
- Finally, with the results of this test, a final version of the WTMST scale was made by the research team to check its suitability for application to Spanish-speaking populations.

Consequently, the instrument used was The Work Tasks Motivation Scale for Teachers (WTMST), which measures teacher motivation. It consists of 15 items measuring five motivational constructs: intrinsic motivation, identified regulation, introjected regulation, external regulation, and demotivation. At the same time, it contemplates six teacher tasks (class preparation, teaching, student evaluation, class management, administrative tasks, and complementary tasks). Therefore, the instrument consists of 90 items, distributed in five motivational constructs assessed in 6 different tasks. Likewise, as in the original scale, we opted for seven degrees of intensity, assigning the value 7 to "corresponds completely", 6 to "corresponds a lot", 5 to "corresponds quite a lot", 4 to "corresponds moderately", 3 to "corresponds a little", 2 to "corresponds very little", and 1 to "does not correspond" (see Figure 2).

**The Work Tasks Motivation Scale for Teachers (WTMST) 15 ITEMS**

**Intrinsic regulation**
Because it is pleasant to carry out this task.
Because I find this task interesting to do.
Because I like doing this task.

**Identified regulation**
Because it is important for me to carry out this task
Because this task allows me to attain work objectives that I consider important
Because I find this task important for the academic success of my students

**Introjected regulation**
Because if I don't carry out this task, I will feel bad
Because I would feel guilty not doing it
To not feel bad if I don't do it.

**External motivation**
Because I'm paid to do it.
Because the school obliges me to do it
Because my work demands it.

**Non-regulation**
I used to know why I was doing this task, but I don't see the reason anymore.
I don't know sometimes I don't see its purpose.
I don't know, I don't always see the relevance of carrying out this task.

**Figure 2.** Items used in the WTMST scale.

### 2.3. Procedure

The present study was conducted during the months of September to December 2019, following the 1975 Declaration of Helsinki. The directors of each center were informed of the objectives of the study, as well as the data collection procedure, and their approval was requested. In addition, voluntary participation in the study was also emphasized and the indications contained in the Organic Law 3/2018, of 5 December 2018, on Personal Data Protection and guarantee of digital rights were followed. The approval of the Ethics Committee of the University of Vigo was also obtained.

### 2.4. Statistical Analysis

This was an instrumental study [48] with a non-experimental cross-sectional survey design, focused on analysis of the metric properties of a measurement instrument: the WTMST scale [27]. The data were analyzed using SPSS Statistics and SPSS Amos software, both version 24. To check the normality of the questionnaire data (Likert-type scale items), the mean, standard deviation, corrected homogeneity index (item-total correlation), skewness, kurtosis, and Mardy coefficient were calculated for each item. Prior to the selection of the extraction method, two indicators of sample adequacy were used: the Kaiser–Meyer–Olkin index (KMO) and Bartlett's test of sphericity. In the first phase of the study with 150 teachers, an exploratory factor analysis (maximum likelihood with Oblimin rotation) was carried out to determine the number of latent factors. Then, the correlation between factors was calculated (Pearson correlation), with the reliability coefficient calculated through Cronbach's alpha and McDonald's omega. In the second phase, with a new sample (N = 219), the validity of three models was tested (confirmatory factor analysis using the generalized least squares method): one-factor, four-factor (achieved), and five-factor (theoretical). In the evaluation of the models, absolute goodness-of-fit measures were used as criteria ($X^2/gl$ ranges from 2.0 to 5.0, GFI $\geq$ 0.90, RMSEA < 0.08, SRMR $\leq$ 0.08) and comparative fit (CFI $\geq$ 0.90) [49–51]. In addition, in order to follow the indications of Byrne (2010) [52], the Aiken information criterion (AIC) was attached. To assess gender invariance, changes in the $X^2$ were taken into account. However, as this

depends on the sample size, it was also considered that variations of the CFI ($\Delta$CFI) $\leq 0.01$, RMSEA ($\Delta$RMSEA) were adequate to accept the invariance.

Finally, in the third phase, after verifying that the data complied with the assumptions of the parametric statistical analyses, a multivariate analysis of variance (MANOVA) was performed. This type of analysis was chosen because it examines the simultaneous effect of multiple variables. In addition, the Wilks' lambda statistic was used, since it is the most commonly used statistic in multivariate analysis when the independent factor under study has more than two treatments. This statistic compares the deviations within each group with the total deviations, without distinguishing groups (significance value: <0.05).

## 3. Results

### 3.1. Preliminary Item Analysis and Mardy Coefficient

Table 1 presents the descriptive statistics, skewness, kurtosis, item-total correlation, and alpha when the item obtained after the application of the questionnaire (N = 369) has been eliminated. Thus, on the one hand, all the items presented values of skewness and kurtosis within the range between $-2$ and $+2$ (outside this range, it is considered that there are indications of non-normality, George & Mallery 2019) [53], adjusted adequately to a univariate normal distribution. In addition, the Mardia coefficient had a value of 46.09 lower than that shown by the proposed equation [p (p + 2)], which indicated multivariate normality [54,55]. On the other hand, the results in the corrected total item correlation were positive in all items ($r_{i-t}$ ranges between 0.250 and 0.668), all contributing to measure the theoretical construct in the same direction. Similarly, the values of the Cronbach's alpha statistic if one item was eliminated (lower than $\alpha_T = 0.825$) showed that the elimination of items would not increase the reliability of the scale.

**Table 1.** Descriptive statistics and Mardía test. Total sample (n = 369).

| ITEMS | MEAN | SD | SKEWNESS (ES = 0.127) | KURTOSIS (ES = 0.253) | $R_{i-t}$ | $\alpha_i$ ($\alpha_T$ = 0.825) |
|---|---|---|---|---|---|---|
| WTMST1 | 24.69 | 7.27 | $-0.02$ | $-0.45$ | 0.385 | 0.818 |
| WTMST2 | 11.56 | 5.72 | 1.35 | 2.16 | 0.238 | 0.825 |
| WTMST3 | 26.66 | 6.92 | $-0.15$ | $-0.21$ | 0.369 | 0.817 |
| WTMST4 | 30.89 | 8.33 | $-0.69$ | $-0.16$ | 0.434 | 0.815 |
| WTMST5 | 35.20 | 5.01 | $-0.84$ | 0.85 | 0.296 | 0.822 |
| WTMST6 | 20.82 | 8.83 | 0.20 | $-0.91$ | 0.594 | 0.803 |
| WTMST7 | 10.04 | 5.30 | 1.57 | 1.97 | 0.293 | 0.822 |
| WTMST8 | 31.60 | 6.38 | $-0.69$ | 0.54 | 0.463 | 0.814 |
| WTMST9 | 29.33 | 6.89 | $-0.46$ | $-0.19$ | 0.391 | 0.817 |
| WTMST10 | 11.30 | 5.51 | 1.31 | 1.72 | 0.250 | 0.829 |
| WTMST11 | 17.89 | 9.92 | 0.51 | $-0.81$ | 0.664 | 0.796 |
| WTMST12 | 17.40 | 9.72 | 0.56 | $-0.73$ | 0.668 | 0.798 |
| WTMST13 | 33.71 | 6.11 | $-1.08$ | 1.76 | 0.276 | 0.823 |
| WTMST14 | 20.39 | 9.80 | 0.15 | $-1.02$ | 0.541 | 0.807 |
| WTMST15 | 15.67 | 9.58 | 0.79 | $-0.38$ | 0.635 | 0.799 |
| MARDIA Coefficient | | 46.09 | [p $\times$ (p + 2), 15 $\times$ (15 + 2) = 255] | | | |

ES: Effect Size.

### 3.2. First Study: Exploratory Factor Analysis of the WTMST Scale

After performing the translation processes and indicating the normality of the data matrix, the first step was to study the construct validity of the scale. Initially, Bartlett's test of sphericity and the Kaiser–Meyer–Olkin index (KMO) were used to determine the appropriateness of the factor analysis. Thus, in this study, Bartlett's test of sphericity (Bartlett's test = 4844.745; *p* < 0.001) and the KMO test (0.831) indicated that the sample taken for the study was appropriate and that the factor analysis could therefore be carried out.

Likewise, after verifying the multivariate normality of the data, the maximum likelihood extraction method with Oblimin rotation was considered suitable for the analysis.

First, extraction was performed with eigenvalues greater than 1, showing the existence of four components (IMR: intrinsic motivation and regulation; YR: introjected regulation; D: demotivation; ER: external regulation) with eigenvalues (4.61, 4.33, 1.83, and 1.30). Second, another extraction was carried out with a fixed number of factors (five), as indicated in the theoretical review (IM: intrinsic motivation; IR: identified regulation; YR: introjected regulation; ER: external regulation; D: demotivation). Next, the extracted component matrix showed the resulting factors for the two models and the questionnaire items included in each (Table 2). Considering the communality (h2), better values were observed in the five-factor forced model (ranging from 0.619 (item 4) to 0.969 (item 12)) than in the four-factor model (ranging from 0.528 (item 5) to 0.961 (item 12)). After the factor analyses and selecting only the matrix items whose saturation was greater than 0.40, Table 2 shows the distribution by item of the four components that explained 80.57% of the variance and the five components that explained 86.20% of the variance.

**Table 2.** Rotated factor structure, eigenvalues, and percentage of variance explained by each factor of the two models (n = 150).

| ITEMS | 4-FACTOR MODEL | | | | 5-FACTOR MODEL | | | | |
|---|---|---|---|---|---|---|---|---|---|
| | YR | IMR | D | ER | YR | IM | IR | D | ER |
| 12. Because if I don't, I will feel bad. | 0.977 | | | | 0.984 | | | | |
| 11. Because I would feel guilty if I didn't do it. | 0.946 | | | | 0.923 | | | | |
| 15. So that I won't feel bad if I don't do it | 0.937 | | | | 0.921 | | | | |
| 3. Because I like doing it | | 0.940 | | | | 0.965 | | | |
| 1. Because it is pleasant to perform this task | | 0.897 | | | | 0.892 | | | |
| 6. Because it is interesting | | 0.821 | | | | 0.613 | | | |
| 7. It is important to me | | 0.784 | | | | | 0.848 | | |
| 8. Because I find this task important for the academic success of my students. | | 0.579 | | | | | 0.814 | | |
| 9. Because this task allows me to achieve goals that I consider important in my work. | | 0.558 | | | | | 0.622 | | |
| 10. I don't know, sometimes I don't see the purpose. | | | 0.908 | | | | | 0.914 | |
| 11. I used to find meaning in this task but not anymore. | | | 0.877 | | | | | 0.894 | |
| 12. I don't know, I don't always find the relevance of this task. | | | 0.876 | | | | | 0.855 | |
| 13. Because my job requires it | | | | 0.791 | | | | | 0.865 |
| 14. Because the center requires it | | | | 0.705 | | | | | 0.808 |
| 15. Because they pay me to do it | | | | 0.694 | | | | | 0.669 |
| 16. AUTOVALUE | 4.61 | 4.33 | 1.83 | 1.30 | 4.61 | 4.33 | 1.83 | 1.30 | 0.84 |
| 17. % VARIANCE | 30.75 | 28.92 | 12.20 | 8.70 | 30.75 | 28.92 | 12.20 | 8.70 | 5.63 |
| 18. % ACCUMULATED VARIANCE | 80.57 | | | | 86.20 | | | | |

Method of extraction: maximum likelihood; method of rotation: Oblimin with Kaiser.

Next (Table 3), to check convergent validity, the correlations (Pearson) between the different dimensions in the two scale models were analyzed. Thus, in the 4-factor model, significant but low correlations were observed (r ranged between 0.160 and 0.440), with a negative correlation between D and IMR (r = −0.282) and no correlation between IMR and YR (r = 0.085). Similarly, the 5-factor model also showed low correlations (r ranged from 0.160 to 0.440), with negative correlations between IM and D (r = −0.192) and IR and D (r = −0.337), and no correlation between IR and YR (r = 0.033).

**Table 3.** Pearson correlations between the components of the WTMST scale (4 and 5 factors).

| 4-MF | 1 | 2 | 3 | 5-MF | 1 | 2 | 3 | 4 |
|---|---|---|---|---|---|---|---|---|
| YR | 1 | | | MI | 1 | | | |
| MRI | 0.085 | 1 | | YR | 0.112 * | 1 | | |
| D | 0.369 ** | −0.282 ** | 1 | ER | 0.104 * | 0.440 ** | 1 | |
| ER | 0.440 ** | 0.185 ** | 0.160 ** | D | −0.192 ** | 0.369 ** | 0.160 ** | 1 |
| | | | | IR | 0.643 ** | 0.033 | 0.248 ** | −0.337 ** |

**. Correlation is significant at the 0.01 level (2-tailed). *. Correlation is significant at the 0.05 level (2-tailed).

Finally, in this first study, reliability was examined through Cronbach's alpha and McDonald's omega for the total sample (N = 369) and for each of the factors, obtaining values that revealed good internal consistency for both models:

1.  Four-Factor Model: $\alpha$ = 0.825, $\omega$ = 0.802 (introjected regulation: $\alpha$ = 0.968, $\omega$ = 0.969; intrinsic motivation and identified regulation: $\alpha$ = 0.903, $\omega$ = 0.906; demotivation: $\alpha$ = 0.919, $\omega$ = 0.920; external regulation: $\alpha$ = 0.836, $\omega$ = 0.843).
2.  Five-Factor Model: $\alpha$ = 0.825, $\omega$ = 0.802 (introjected regulation: $\alpha$ = 0.968, $\omega$ = 0.969; intrinsic motivation: $\alpha$ = 0.916, $\omega$ = 0.921; identified regulation: $\alpha$ = 0.865, $\omega$ = 0.868; demotivation: $\alpha$ = 0.847, $\omega$ = 0.848; external regulation: $\alpha$ = 0.847, $\omega$ = 0.848).

*3.3. Second Study: Confirmatory Factor Analysis of the WTMST Scale*

According to the results of the exploratory factor analysis, two measurement models (four and five factors) were proposed for "task performance by teachers in their teaching work" (see Figure 3).

When specifying the two models, very high standardized saturations were observed for all items (values above 0.50 indicated by Byrne, 2010) [52], with better item regression weights in the 5-factor model (ranging from 0.72 (item 4) to 0.98 (item 12)) than in the 4-factor model (ranging from 0.58 (item 13) to 0.98 (item 12)). The items that best defined the tasks in the performance of the teaching job were item 12 (D: "Because if I don't do it, I will feel bad") and item 3 (IM: "Because I like to do it"). Likewise, the correlation between the factors was low–moderate and positive, ranging between r = 0.10 (IR-YR) and r = 0.67 (IM-IR). Note here the low and significant negative correlation of the two models between the variables intrinsic motivation (regulation) and demotivation (r = −0.12).

Once the most suitable structural models were found, the most suitable one was compared and selected. Thus, the fit of the different models for the WTMST scale (achieved (4-MF), theoretical (5-MF), unidimensional (1-MF)) was tested. The one-dimensional model exhibited inadequate values; however, the original and the revised model produced an adequate fit. Nevertheless, the model forced to use five factors (theoretical) proved to fit the calibration sample better ($\chi^2$/gl = 5.09; CFI = 0.932; TLI = 0.911; GFI = 0.902; SRMR = 0.067; RMSEA = 0.077). Likewise, the AIC value (478.81) of the theoretical model was smaller than that achieved with the four-factor model (AIC = 684.24) (Table 4).

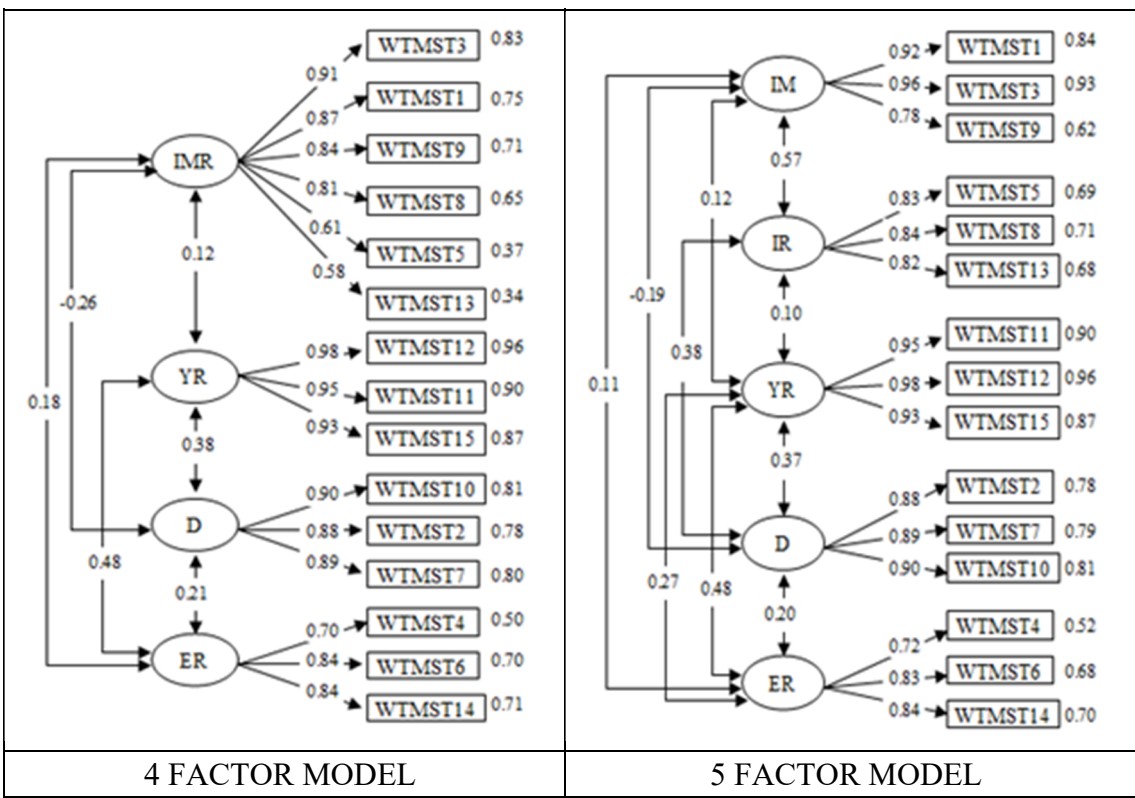

**Figure 3.** Standardized regression coefficients and standardized factor saturations of the WTMST scale (4 factors, 5 factors).

**Table 4.** Fit indices of proposed models (n = 220).

| Models | $\chi^2_{(df)}$ | $\chi^2/df$ | CFI | TLI | GFI | SRMR | RMSEA (CI) | AIC |
|---|---|---|---|---|---|---|---|---|
| Model achieved WTMST | 713.35 (84) | 8.49 | 0.896 | 0.873 | 0.897 | 0.089 | 0.143 (0.133–0.152) | 684.24 |
| Theoretical model WTMST | 407.81 (80) | 5.09 | 0.932 | 0.911 | 0.902 | 0.067 | 0.077 (0.065–0.089) | 478.81 |
| One-dimensional model WTMST | 3473.7 (90) | 38.59 | 0.498 | 0.281 | 0.512 | 0.265 | 0.320 (0.311–0.329) | 3533.7 |

Note: $\chi^2$: Chi-square; gl: degrees of freedom; RMSEA: root mean square error of approximation; CI: confidence intervals; SRMR: standardized root mean squared residual; CFI: comparative fit index; GFI: goodness of fit index; AIC: Aiken information criterion.

As for the five-factor model, gender invariance was examined for the whole sample (n = 369). As presented in Table 5, the configural invariance showed an acceptable fit with the data ($\chi^2/df$ = 6.62; CFI = 0.924; TLI = 0.903; SRMR = 0.068; RMSEA = 0.082). We subsequently tested for metric invariance ($\chi^2/df$ = 6.51; CFI = 0.920; TLI = 0.907; SRMR = 0.080; RMSEA = 0.079) and scalar invariance ($\chi^2/df$ = 6.36; CFI = 0.913; TLI = 0.910; SRMR = 0.083; RMSEA = 0.079), being invariant for gender (ΔCFI = 0.007; ΔRMSEA < 0.001).

**Table 5.** Gender invariance: summary of goodness-of-fit indices (N = 369).

| Invariance Models | $X^2$ | df | CFI | TLI | RMSEA | RMSEA 90% CI | SRMR | ΔCFI | ΔRMSEA |
|---|---|---|---|---|---|---|---|---|---|
| 1. Configural | 987.18 | 149 | 0.924 | 0.903 | 0.082 | 0.059–0.106 | 0.068 | - | - |
| 2. Metric | 1002.104 | 154 | 0.920 | 0.907 | 0.079 | 0.058–0.104 | 0.080 | 0.004 | 0.003 |
| 3. Scalar | 1017.701 | 160 | 0.913 | 0.910 | 0.079 | 0.059–0.101 | 0.083 | 0.007 | <0.001 |

### 3.4. Third Study: Relationship of Gender and Age with the Five Factors of the WTMST Questionnaire

In descriptive terms, female gender obtained higher mean scores in two WTMST questionnaire factors (D, RI), while male gender obtained higher mean scores in three factors (RY, RE and MI). Likewise, multivariate MANOVA analysis demonstrated a significant influence of gender ($\lambda$Wilks = 0.94, F(363) = 4.39, $p < 0.01$, $\eta p^2 = 0.06$), although it was only statistically significant for identified regulation [F(1367) = 7.56, $p = 0.006$, $\eta p^2 = 0.020$], with better regulation identified in the male gender.

The effect of age was also significant ($\lambda$Wilks = 0.88, F(362) = 4.65, $p < 0.001$, $\eta p^2 = 0.06$). Thus, significant differences were shown for introjected regulation [F(2366) = 4.27, $p = 0.015$, $\eta p^2 = 0.023$], demotivation [F(2366) = 7.49, $p = 0.001$, $\eta p^2 = 0.039$], intrinsic motivation [F(2366) = 7.77, $p = 0.000$, $\eta p^2 = 0.041$], and identified regulation [F(2366) = 5.56, $p = 0.004$, $\eta p^2 = 0.029$], but not external regulation [F(2366) = 0.007, $p = 0.993$, $\eta p^2 = 0.000$]. The a posteriori analyses (Tukey's DHS) indicated greater introjected regulation in teachers older than 52 years versus those younger than 38 years, and greater demotivation in teachers older than 52 years versus those younger than 38 years and those between 38 and 52 years (Table 6).

**Table 6.** Means, standard deviations of gender and age according to the WTMST.

| GENDER | YR | D | ER | IM | IR |
|---|---|---|---|---|---|
| Male (n = 106) | 51.57 (6.11) | 32.24 (2.78) | 72.12 (4.18) | 81.52 (9.98) | 96.02 (7.28) |
| Female (n = 263) | 50.72 (9.24) | 33.17 (3.29) | 72.11 (3.21) | 80.34 (9.36) | 99.82 (4.68) |
| **AGE** | **YR** | **D** | **ER** | **IM** | **IR** |
| <38 years (n = 99) | 44.95 (7.59) | 29.12 (3.79) | 72.02 (3.76) | 87.18 (7.63) | 99.43 (6.87) |
| 38–52 years (n = 180) | 51.32 (8.35) | 32.65 (5.26) | 72.26 (3.62) | 78.42 (8.64) | 93.05 (8.04) |
| >52 years (n = 90) | 56.87 (8.14) | 37.60 (8.43) | 71.92 (3.07) | 78.06 (9.67) | 96.13 (8.67) |

Note: IM: intrinsic motivation; IR: identified regulation; YR: introjected regulation; ER: external regulation; D: demotivation.

### 4. Discussion

Work motivation is one of the most important constructs in psychology, being widely studied by academics and practitioners [56]. As Lusková & Hudáková (2015) [57] explained, the administrations of all organizations have to cope with the task of recruiting and retaining competent staff, especially by achieving adequate motivation in their workers. Motivation, as one of the basic preconditions of effective and successful performance of employees at work, is also an essential part of human resource management in schools. Well-motivated teachers are people with clearly defined goals and who take action to achieve them. They have developed a strong sense of duty and responsibility. Having a valid and reliable tool that allows us to determine the level of teacher motivation seems to be of utmost importance. Therefore, the aim of this study was to analyze the psychometric properties of the Spanish translation of the WTMST scale in a sample of teachers in the Autonomous Community of Galicia (Spain), in order to determine the reliability and validity of this scale for Spanish-speaking populations. The statistical analyses carried out for this purpose indicated that the WTMST [27] in Spanish has adequate psychometric properties to be used in the Spanish teaching population, since it yielded reliability and validity coefficients in accordance with those required by the scientific literature [46,49,58–63].

At first, the steps indicated for the adaptation of scales to another language were followed, achieving a consensual and homologous version, both conceptually and linguistically, with the original version. Furthermore, the results obtained support that the psychometric properties of the WTMST scale in Spanish are consistent with the original English version [27]. This affirmation was performed by corroborating the metric quality of the items that make up the scale, its factorial structure, and the internal consistency of the factors that form it. Thus, the discrimination capacity of the items through the homogeneity index (total-corrected item correlation) were considered acceptable (not lower than 2),

determining the deletion of no items [64–66]. This was supported by the Cronbach's Alpha if an item was deleted, since none exceeded that achieved for the total scale and their deletion would not improve the test–retest reliability.

Likewise, the mean value of the items in the scale presented higher values for intrinsic motivation ("Because I find this task important for the academic success of my students"; "Because this task allows me to achieve goals that I consider important"; "It is important to me"), and the lowest were related to demotivation ("Previously I found meaning in this task but not anymore"; "I don't know, sometimes I don't see the purpose"; "I don't know, I don't always find the relevance of this task").

Then, as a prior step to the factor analysis, it was verified that the data achieved for skewness and kurtosis remained within the interval considered acceptable and leading to normality of the distribution in the scores [67,68]. This was complemented with Mardia's test, which showed the multivariate normality of the data [55]. Next, the data obtained in Bartlett's test of sphericity and the Kaiser–Meyer–Olkin index (KMO) indicated that such a factor analysis could be carried out.

Taking gender into account, there was greater demotivation and identified regulation in women and greater introjected regulation, external regulation, and intrinsic motivation in men. However, significance was only shown for identified regulation, being higher in the male gender. According to age, significant differences were found in introjected regulation, demotivation, intrinsic motivation, and identified regulation. Thus, older teachers reported better introjected regulation and being more demotivated, while younger teachers reported better intrinsic motivation and identified regulation.

Furthermore, our results for gender invariance suggested that the WTMST scale was equally valid for women and men.

Thus, with regard to the construct validity evaluated (exploratory factor analysis), a four-factor factor structure was found (intrinsic motivation and regulation, introjected regulation, demotivation, and external regulation), which does not agree with the original model (five factors: intrinsic motivation, identified regulation, introjected regulation, demotivation, and external regulation), although it presented a good sample adequacy and an explained common variance of 80.57% (exceeding the quality criterion of at least 50% explained covariance) and with adequate factorial weights in its items. However, it was decided to also test the (original) five-factor model through an exploratory analysis forced to use a fixed extraction of five factors. The data obtained also showed a good sample adequacy and its components explained 86.20% of the variance (a slight increase of 5.63% in the accumulated variance with respect to the four-factor model) and with high factorial weights in its items.

Consequently, in order to contrast the factorial structure and verify the best fit, a confirmatory factor analysis was carried out by testing three models: unidimensional, achieved (four factors), and theoretical (five factors). The data supported an adequate fit in the achieved and theoretical models, in addition to the inadequacy of the unidimensional model. However, although the fit of the two models studied was acceptable, the theoretical model (five factors) was slightly better by presenting higher values for its factor structure, explained variance, regression coefficients, standardized factor saturations, and incremental fit indices (CFI, TLI, GFI), and lower values for the absolute indices (SRMR, RMSEA) and AIC.

It should also be noted that both models presented moderate and statistically significant correlations between their factors. The negative correlation between demotivation and intrinsic motivation or identified regulation should be highlighted here, which shows that better intrinsic motivation or identified regulation decrease demotivation in teachers. In addition, the reliability coefficients (Cronbach's alpha and McDonald's omega) were good, both in the total scales and in the factors that comprised them, with estimates in line with those recommended by [53,69].

## 5. Limitations and Educational Implications

Thus, the results of the study indicated that the Spanish version of the WTMST is a valid and reliable instrument for measuring teacher motivation, and they confirmed that the theoretical factor structure (five factors: introjected regulation, intrinsic motivation, identified regulation, demotivation, and external regulation) best fits the measurement model for the six tasks that affect their quality of work (class preparation, teaching, student assessment, classroom management, administrative tasks, and complementary tasks).

Teacher motivation is vital for the education system. For teachers to be motivated, their job satisfaction and positive psychological capital are crucial [70]. If we would like to develop a program to improve the motivation of our teachers, it would be necessary to have a basic idea about its form and concrete implications in the centers.

If we focus on their daily tasks, knowing the teacher's motivation during each of them has various educational implications that could significantly influence student performance and the learning environment. Understanding teacher motivations is essential to promoting a healthy and effective educational environment. This involves not only recognizing individual motivations, but also creating conditions that foster and maintain high levels of motivation over time. Likewise, knowing the motivation of teachers during the different tasks to which they dedicate their daily lives could have important repercussions on the quality of the educational system. More specifically, knowing the motivation of teachers when preparing their classes not only benefits teachers, but also has a direct impact on the learning experience of students. Motivated teachers tend to create a more dynamic, inspiring, and effective learning environment. If we look at their motivation when assessing their students, this can lead to a more holistic assessment process, focused on student development and promoting meaningful learning. This contributes to an educational environment that values each student's growth and progress.

Faculty motivation when performing administrative tasks can have a significant impact on the operational efficiency of the educational institution, creating an environment conducive to student learning and development. Understanding these motivations can guide strategies to optimize school management and improve the quality of education. In addition, understanding teacher motivation in classroom management not only improves the learning experience of students, but also contributes to the formation of more engaged and motivated citizens. Motivated teachers create a dynamic and stimulating classroom environment that favors the holistic development of students, and they tend to opt for more inclusive teaching styles, promote active learning, recognize the specific strengths and challenges of each student, use positive and constructive approaches to discipline management, and promote an environment of mutual respect.

To conclude, motivational programs in educational centers should focus exclusively on the positive influence on the work motivation of teachers, and for this we consider it necessary to have a tool that can give us an accurate picture of the motivation of teachers in the center, to focus the program on what is really failing. This is where the scale presented in this article stands out from the rest, since it is the only one that focuses on the teacher's tasks. Knowing not only the type of motivation experienced by the teachers, but more specifically the tasks that produce this day-to-day wear and tear and that are the source of their demotivation, is an excellent starting point if what we seek is to improve the quality of our centers and the well-being of the teaching staff. In addition, it should be noted that, depending on requirements, we could use smaller versions of the survey, adapting it to the needs of the study.

As limitations, it would be interesting to extend the study sample to other autonomous communities and to complement this self-reported measure with the use of other qualitative techniques (interviews, focus groups, and external assessments). In addition, comparisons were only made with the one-factor model and the theoretical model, so it would be interesting to make further comparisons with other factor models.

In view of the satisfactory results achieved and the suspicion of a decrease in teacher motivation, the use of Spanish adaptations of scales to determine the motivational level

of teachers could contribute to improving their teaching performance, through screening, evaluation, intervention, and prevention.

**Author Contributions:** Conceptualization, J.D.-A. and J.C.-D.R.; methodology, J.D.-A. and M.P.-J., formal analysis, J.D.-A. and M.P.-J.; writing, I.P.-P. and J.C.-D.R.; review and editing, M.P.-J. and J.C.-D.R. All authors have read and agreed to the published version of the manuscript.

**Funding:** This research received no external funding.

**Institutional Review Board Statement:** The study was conducted in accordance with the Declaration of Helsinki. Given the nature of the study, it was not necessary to pass an ethics committee.

**Informed Consent Statement:** Informed consent was obtained from all subjects involved in the study.

**Data Availability Statement:** The data presented in this study are available on request from the corresponding author.

**Acknowledgments:** The authors would like to thank the reviewers for their feedback and input towards improving the quality of this paper.

**Conflicts of Interest:** There were no personal or financial conflicts of interest that could have caused bias of the contents of this paper.

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
