# Peer review of "Assessment of Teacher Motivation, Psychometric Properties of the Work Tasks Motivation Scale for Teachers (WTMST) in Spanish Teachers"

_education, doi:10.3390/educsci14030212_

Round 1

Reviewer 1 Report

Comments and Suggestions for Authors

Abstract should make results in context rather than statistical results.

The topic is should be further elaborated and explained in the literature review. What is the significance of the topic? What had been done? More background of the study in the context of Spanish teachers should be given more attention than how it is stated in the paper now.

Also, how were the hypotheses developed based on previous studies or literature? 

The section on methodology should also be strengthened. The form of presentation is now rather confusing. I suggest this order: Design, procedures, instruments, participants, data analysis...etc. 

The results were well explained; however, what is the significance? What new knowledge has brought to the field? What implications can be drawn based on the findings? These are important for conducting research. 

A conclusion of the study is missing. 

I believe this paper has its merits in terms of results. However, it must be improved before it can be published. 

Comments on the Quality of English Language

The language in this paper is generally fine, but the structure can be improved. 

Author Response

Dear reviewer, 

Thank you very much for your suggestions and comments. I will proceed to answer each of your concerns.

Abstract should make results in context rather than statistical results. Done

The topic is should be further elaborated and explained in the literature review. What is the significance of the topic? What had been done? More background of the study in the context of Spanish teachers should be given more attention than how it is stated in the paper now.

It has been expanded with a paragraph on the importance of the subject by adding updated citations and international databases.

Also, how were the hypotheses developed based on previous studies or literature? 

The reliability and validity of the original scale have been added. Other instruments are also described in the theoretical framework to support our hypothesis.

The section on methodology should also be strengthened. The form of presentation is now rather confusing. I suggest this order: Design, procedures, instruments, participants, data analysis...etc. 

The scheme used to describe the methodology is the usual one in education studies and also the format used in this journal.

The results were well explained; however, what is the significance? What new knowledge has brought to the field? What implications can be drawn based on the findings? These are important for conducting research. 

The implications of the study and the importance of the subject matter are described in the discussion section and focus on :

Thus, the results of the study indicate that the Spanish version of the WTMST is a valid and reliable instrument for measuring teacher motivation, and confirm that the theoretical factor structure (five factors: introjected regulation, intrinsic motivation, identified regulation, demotivation and external regulation) best fits the measurement model for the six tasks that affect their quality of work (class preparation, teaching, student assessment, classroom management, administrative tasks and complementary tasks).

Teacher motivation is vital to the education system. For teachers to be motivated, their job satisfaction and positive psychological capital are crucial (Viseu et al., 2016). If we would like to develop a program to improve the motivation of our teachers, it would be necessary to have a basic idea about its form and concrete implications in the centers.

A conclusion of the study is missing. 

A sentence has been added with the final conclusion of the study.

I believe this paper has its merits in terms of results. However, it must be improved before it can be published. 

Reviewer 2 Report

Comments and Suggestions for Authors

Thank you for giving me this opportunity to review this manuscript titled 'Assessment of teacher motivation. Psychometric properties of the work tasks motivation scale for teachers (WTMST) in spanish teachers.' My suggestions are detailed as below.

Introduction

1. In Lines 34-36 'Being a teacher involves the personal and professional life of the teacher, which is why it is so important to take care of the teacher's motivation and job satisfaction.'  This statement leaves the readers wonder why personal and professional life of teachers can justify the importance of investigating teacher motivation. Perhaps authors can elaborate a bit more past research findings about the consequences of lack of teacher motivation in personal (such as stress and burnout) and professional life (low work performance). Also, all the cited articles [11-14] are in Spanish. Perhaps authors can supplement several research findings from English journals as well, so international audience would have more ease to read relevant findings if interested.

2. In Lines 62-68 authors provided only a thin overview of domain specificity in assessing teachers' motivation. It is unclear how these domains are related to one another. As the findings indicated low correlations between the domains/factors, the authors should first elaborate the definition of each domain and summarize the past research findings (especially Fernet et al. 2008) for a better understanding of each construct/domain.  

3. Lines 81-99  The review for some instruments in prior studies may not be necessary, as efficacy aspects and goal theory are not relevant to the adapted WTMST instrument.

Method

3. Lines 153-158 Could authors please specify the total number of  items in the instrument (which should be 15 items for 6 tasks, a total of 90 items). Please also articulate how the level of each motivation was assessed for each task type. 

Statistical Analysis

4. Given that this instrument has a total of 90 items, authors should articulate how many latent factors were estimated (e.g., 5 constructs x 6 task types). It appears that authors ignored the task type and simply averaged the scores across task types to create a composite score for each motivation construct. 

5. Relatedly, authors omitted a test of 5-factor structure in relation to each task type. As the authors argued that teachers exhibited different motivation while engaging in different tasks, authors should estimate a 30-factor structure (5 factors for 6 task types; see Fernet et al., 2008). This test is crucial because it provides clear evidence for convergent and discriminant validity (such as intrinsic motivation in relation to class preparation and amotivation in relation to admin tasks).

6. Lines 181-188  The sample of Study 1 was used to conduct a EPA, whereas that of Study 2 was used for a CFA. I suggest that authors combine the both samples to increase the power.

7. It is highly recommended to conduct multi-group invariance test (by gender or teaching experience) so as to provide a more stringent test of the construct validity. 

Results

8. Tables 1-3:  Please attach the results for all 90 items or provide a justification if facets have been created from item scores. 

Discussion

9. Lines 323-326  Similar to point (4), it's inappropriate to compare the current findings about the factor structure to past research findings, as authors did not carry out CFA for all 90 items. 

Author Response

Dear reviewer,

Thank you very much for your comments and suggestions for improvement. I will proceed to answer each of your concerns.

  1. In Lines 34-36 'Being a teacher involves the personal and professional life of the teacher, which is why it is so important to take care of the teacher's motivation and job satisfaction.'  This statement leaves the readers wonder why personal and professional life of teachers can justify the importance of investigating teacher motivation. Perhaps authors can elaborate a bit more past research findings about the consequences of lack of teacher motivation in personal (such as stress and burnout) and professional life (low work performance). Also, all the cited articles [11-14] are in Spanish. Perhaps authors can supplement several research findings from English journals as well, so international audience would have more ease to read relevant findings if interested.

This idea has been expanded in the text

  1. In Lines 62-68 authors provided only a thin overview of domain specificity in assessing teachers' motivation. It is unclear how these domains are related to one another. As the findings indicated low correlations between the domains/factors, the authors should first elaborate the definition of each domain and summarize the past research findings (especially Fernet et al. 2008) for a better understanding of each construct/domain.  

A paragraph has been added with a theoretical approach to the constructs analyzed.

  1. Lines 81-99  The review for some instruments in prior studies may not be necessary, as efficacy aspects and goal theory are not relevant to the adapted WTMST instrument.

The review of the instruments used to measure teacher motivation has allowed us to develop the hypothesis formulated. It is precisely another reviewer who asked us to add more information on these instruments. We have opted to keep the paragraph and add reliability and validity data on the original scale

Method

  1. Lines 153-158 Could authors please specify the total number of  items in the instrument (which should be 15 items for 6 tasks, a total of 90 items). Please also articulate howthe level of each motivation was assessed for each task type.  A sentence was added to make the organization of the scale more understandable, which is detailed in Figure 1.

Statistical Analysis

  1. Given that this instrument has a total of 90 items, authors should articulate how many latent factors were estimated (e.g., 5 constructs x 6 task types). It appears that authors ignored the task type and simply averaged the scores across task types to create a composite score for each motivation construct. 

Indeed, the scores were averaged across task types in order to then analyze the psychometric properties of the total scale, not of the individual teacher tasks.

  1. Relatedly, authors omitted a test of 5-factor structure in relation to each task type. As the authors argued that teachers exhibited different motivation while engaging in different tasks, authors should estimate a 30-factor structure (5 factors for 6 task types; see Fernet et al., 2008). This test is crucial because it provides clear evidence for convergent and discriminant validity (such as intrinsic motivation in relation to class preparation and amotivation in relation to admin tasks).

The aim of the study is to validate the instrument to measure the motivational level of the teacher. It is not the intention of this article to determine the reliability of each of the items linked to each task. We thank the reviewer for the suggestion and confirm that the reliability of each task and its association with descriptive variables will be described in another publication, currently under review. We have to take into account the complexity of a 30-factor model to reflect all the information in the same article. 

At first, all the tasks have been combined in the same item (for example: in item 1, the responses to the item have been combined: 1class preparation + 1teaching + 1student assessment + 1class management + 1administrative tasks + 1additional tasks; in item 2, the responses to the item have been combined: 2class preparation + 2administrative tasks + 1additional tasks; in item 2, the responses to the item have been combined: 2class preparation + 2administrative tasks + 2additional tasks + 2additional tasks: 2classroom preparation + 2teaching + 2student assessment + 2classroom management + 2administrative tasks + 2additional tasks; and so on until item 15. The reliability and validity of the questionnaire was then calculated.

  1. Lines 181-188  The sample of Study 1 was used to conduct a EPA, whereas that of Study 2 was used for a CFA. I suggest that authors combine the both samples to increase the power.

The usual scheme for this type of study was followed. First, a small sample was used to determine the factorial structure, and once this was confirmed, the confirmatory analysis was carried out.

  1. It is highly recommended to conduct multi-group invariance test (by gender or teaching experience) so as to provide a more stringent test of the construct validity. 

Another article under review includes descriptive analyses using Mancova by gender and teaching experience. Given the complexity of the study, it has been decided to carry out two different articles that allow a better understanding of the tool. The objective of this study is to confirm its use in teacher motivation and not so much the descriptive detail of each of the factors.

Results

  1. Tables 1-3:  Please attach the results for all 90 items or provide a justification if facets have been created from item scores. These data are the subject of another article.

Discussion

  1. Lines 323-326  Similar to point (4), it's inappropriate to compare the current findings about the factor structure to past research findings, as authors did not carry out CFA for all 90 items. 

The study did perform the factorial structure with the 90 items, calculating the average of each of the factors in all the tasks. The differences are calculated by gender and age.

Reviewer 3 Report

Comments and Suggestions for Authors

Dear Editor,

I would like to express my gratitude to you and the Associate Editor for giving me the opportunity to review this manuscript. I approached the article with great enthusiasm and believe that it offers valuable insights into the psychometric properties of the Spanish translation of the Work Tasks Motivation Scale for Teachers (WTMST) in a sample of teachers from the Autonomous Community of Galicia, Spain. The study's focus on determining the reliability and validity of the scale for Spanish-speaking populations is commendable. Additionally, the manuscript has the potential to contribute to the cultural context of Spain and Spanish-speaking populations, with the authors' recommendations, opening the door for international cross-cultural validation.

Regarding the specific aspects of the article:

Article Title:

·       The article's title, "Assessment of Teacher Motivation: Psychometric Properties of the Work Tasks Motivation Scale for Teachers (WTMST) in Spanish Teachers," is relevant and aligns well with the aims and scope of the journal.

·       The title effectively reflects the content of the manuscript, and the topic itself is original and meaningful to the study.

The Literature:

·       The literature review in the manuscript is sound, timely, and well-articulated.

·       Previous models have been compared, and the study clearly identifies the research gap that it aims to address.

Methodology:

·       The statistical analyses conducted in the study demonstrate that the Spanish version of the WTMST exhibits adequate psychometric properties, including reliability and validity coefficients that align with the requirements of scientific literature.

·       The adaptation process followed recommended steps for translating scales to another language, resulting in a conceptually and linguistically equivalent version of the original scale.

·       However, I have a concern regarding the sample population used for the Exploratory Factor Analysis (EFA) and Confirmatory Factor Analysis (CFA). It would greatly benefit the study if the adequacy of the chosen sample population is supported by evidence or existing literature. Justification for the selection would enhance the study's validity.

·       While EFA and CFA are appropriate for establishing the psychometric properties of the WTMST, I suggest including a limitation on model comparison, such as single factor, bi-factor model, higher-order model, and correlated factor model. The absence of these models in the limitation section could be mentioned as a potential area for future research.

·       Measurement equivalence in cross-cultural validation is crucial. I recommend including the measurement of the WTMST across gender, age, or demographic factors, or providing a rationale for not doing so.

Findings:

The study effectively emphasizes the importance of teacher motivation in the education system and the necessity of a reliable tool for accurate assessment. The WTMST scale, with its focus on teacher tasks, is deemed valuable for understanding not only the types of motivation experienced by teachers but also the specific tasks that contribute to demotivation. This information holds potential for the development of programs aimed at improving teacher motivation and well-being.

In conclusion, this manuscript presents a well-structured review of the psychometric properties of the Spanish translation of the WTMST. It addresses important aspects of teacher motivation and provides valuable insights. However, I recommend addressing the concerns raised about sample population justification, model comparison, and measurement equivalence. Overall, this study has the potential to contribute significantly to the field and warrants consideration for publication.

Thank you once again for the opportunity to review this work.

References

Davidov, E., Schmidt, P., Billiet, J., & Meuleman, B. (2018). Cross-cultural analysis:Methods and application: Vol. 2nd Editio. Taylor & Francis Group, LLC.

Millsap, R. E. (2011). Stastical approaches to measurement invariance. Taylor and Francis Group, LLC.

Author Response

Dear Editor,

I would like to express my gratitude to you and the Associate Editor for giving me the opportunity to review this manuscript. I approached the article with great enthusiasm and believe that it offers valuable insights into the psychometric properties of the Spanish translation of the Work Tasks Motivation Scale for Teachers (WTMST) in a sample of teachers from the Autonomous Community of Galicia, Spain. The study's focus on determining the reliability and validity of the scale for Spanish-speaking populations is commendable. Additionally, the manuscript has the potential to contribute to the cultural context of Spain and Spanish-speaking populations, with the authors' recommendations, opening the door for international cross-cultural validation.

Thank you very much for your feedback, we really appreciate it.

Regarding the specific aspects of the article:

Article Title:

  • The article's title, "Assessment of Teacher Motivation: Psychometric Properties of the Work Tasks Motivation Scale for Teachers (WTMST) in Spanish Teachers," is relevant and aligns well with the aims and scope of the journal.
  • The title effectively reflects the content of the manuscript, and the topic itself is original and meaningful to the study.

The Literature:

  • The literature review in the manuscript is sound, timely, and well-articulated.
  • Previous models have been compared, and the study clearly identifies the research gap that it aims to address.

Methodology:

  • The statistical analyses conducted in the study demonstrate that the Spanish version of the WTMST exhibits adequate psychometric properties, including reliability and validity coefficients that align with the requirements of scientific literature.
  • The adaptation process followed recommended steps for translating scales to another language, resulting in a conceptually and linguistically equivalent version of the original scale.
  • However, I have a concern regarding the sample population used for the Exploratory Factor Analysis (EFA) and Confirmatory Factor Analysis (CFA). It would greatly benefit the study if the adequacy of the chosen sample population is supported by evidence or existing literature. Justification for the selection would enhance the study's validity.

In the selection of the sample, we have taken into account that it should be complete for the purposes of the objectives pursued, representative of the total teaching staff and reliable (in terms of the goodness of the teachers selected and their suitability).

  • While EFA and CFA are appropriate for establishing the psychometric properties of the WTMST, I suggest including a limitation on model comparison, such as single factor, bi-factor model, higher-order model, and correlated factor model. The absence of these models in the limitation section could be mentioned as a potential area for future research.

Limitation - comparisons were only made with the one-factor model and the theoretical model, so it would be interesting to make further comparisons with other factor models.

  • Measurement equivalence in cross-cultural validation is crucial. I recommend including the measurement of the WTMST across gender, age, or demographic factors, or providing a rationale for not doing so.

Measurement by age and gender is included.

Findings:

The study effectively emphasizes the importance of teacher motivation in the education system and the necessity of a reliable tool for accurate assessment. The WTMST scale, with its focus on teacher tasks, is deemed valuable for understanding not only the types of motivation experienced by teachers but also the specific tasks that contribute to demotivation. This information holds potential for the development of programs aimed at improving teacher motivation and well-being.

In conclusion, this manuscript presents a well-structured review of the psychometric properties of the Spanish translation of the WTMST. It addresses important aspects of teacher motivation and provides valuable insights. However, I recommend addressing the concerns raised about sample population justification, model comparison, and measurement equivalence. Overall, this study has the potential to contribute significantly to the field and warrants consideration for publication.

Thank you once again for the opportunity to review this work.

References

Davidov, E., Schmidt, P., Billiet, J., & Meuleman, B. (2018). Cross-cultural analysis:Methods and application: Vol. 2nd Editio. Taylor & Francis Group, LLC.

Millsap, R. E. (2011). Stastical approaches to measurement invariance. Taylor and Francis Group, LLC.

Round 2

Reviewer 1 Report

Comments and Suggestions for Authors

I believe the paper has its value to the field. To further enhance its significance, you may consider:

1. Discuss the results with significance to education 

2. Add a section of implication. What do the study's results imply?

3. What is the limitation of the study? 

4. What is the direction of future study? 

Author Response

Comments and Suggestions for Author 1

I believe the paper has its value to the field. To further enhance its significance, you may consider:

  1. Discuss the results with significance to education 
  2. Add a section of implication. What do the study's results imply?
  3. What is the limitation of the study? 
  4. What is the direction of future study? 

A section on limitations and educational implications has been specified in view of the reviewer's comments.

In addition, we have expanded the discussion on the importance of knowing the motivation of teachers in their daily tasks and the impact that this motivation can have on the teaching-learning process.

As for the future direction of our research, it is none other than to continue to learn about the motivation of our teachers, and to be able to invest in concrete resources to improve the educational system. It is considered interesting to extend the study sample to other autonomous communities and to complement this self-reported measure with the use of other qualitative techniques (interviews, focus groups, external assessments). In addition, comparisons were only made with the one-factor model and the theoretical model, so it would be interesting to make further comparisons with other factor models.

Reviewer 2 Report

Comments and Suggestions for Authors

I am happy to review the present article again and would like to express my appreciation to the author(s) for incorporating the prior suggestions.  I think the manuscript has improved since the previous round, yet have identified several outstanding concerns about statistical analysis that would need to be addressed. 

1. The major issue for the current version is the lack of multi-group invariance test.  As I have pointed out (see Comment 7 in the previous round of reviews),  an invariance test is essential to examine whether teachers of different genders have different response patterns.  Authors should still carry out multi-group invariance test for teachers' gender, which will provide a more rigorous test for construct validity of the instrument.    

2. The newly added section (pp. 12-13) tells us only observed means on the subscales differ by gender.  Given the main purpose of this work being to analyze the psychometric properties of the instrument, an invariance test will help address whether the factor structure differs with respondent gender.  I urge authors to include invariance tests with respect to both gender and age. 

Minor issues

In Line 317, It is inappropriate to number the multivariate analysis as Study 3, as authors did not list the information about Study 3 in the Methodology or any preceding sections.

Author Response

Comments and Suggestions for Author 2

I am happy to review the present article again and would like to express my appreciation to the author(s) for incorporating the prior suggestions.  I think the manuscript has improved since the previous round, yet have identified several outstanding concerns about statistical analysis that would need to be addressed. 

  1. The major issue for the current version is the lack of multi-group invariance test.  As I have pointed out (see Comment 7 in the previous round of reviews),  an invariance test is essential to examine whether teachers of different genders have different response patterns.  Authors should still carry out multi-group invariance test for teachers' gender, which will provide a more rigorous test for construct validity of the instrument.    
  2. The newly added section (pp. 12-13) tells us only observed means on the subscales differ by gender.  Given the main purpose of this work being to analyze the psychometric properties of the instrument, an invariance test will help address whether the factor structure differs with respondent gender.  I urge authors to include invariance tests with respect to both gender and age. 

The suggested statistical calculations have been carried out.

Minor issues

In Line 317, It is inappropriate to number the multivariate analysis as Study 3, as authors did not list the information about Study 3 in the Methodology or any preceding sections.

We have proceeded to cite in study 3 in the Statistical Analysis section.